# REVO-LION: Evaluating and Refining Vision-Language Instruction Tuning Datasets

## Abstract

There is an emerging line of research on multimodal instruction tuning, and various benchmarks have been proposed for evaluating these models correspondingly. Instead of evaluating the models directly, in this paper we try to evaluate the Vision-Language Instruction-Tuning (VLIT) datasets themselves and even seek the way of building a dataset for developing an all-powerful VLIT model, which we believe could be fundamental for establishing a grounded protocol for benchmarking VLIT models. To achieve effective analysis of VLIT datasets, which remains an open question, we propose a *tune-cross-evaluation* paradigm: tuning on one dataset and evaluating on others in turn. For each tune-evaluation set, we define the Meta Quality (MQ) as the mean score measured by BLEU, METEOR, and ROUGE-L to quantify the quality of a dataset or a sample. On this basis, to evaluate the comprehensiveness of a dataset, we develop the Dataset Quality (DQ) covering all tune-evaluation sets. To lay the foundation for building a comprehensive dataset and developing an all-powerful model, we further create the Sample Quality (SQ) quantifying the all-sided quality of each sample. Extensive experiments validate the rationality of the proposed evaluation paradigm. According to the holistic evaluation, we build a new dataset, REVO-LION (REfining VisiOn-Language InstructiOn tuNing), by collecting samples with higher SQ from each dataset. With only half of the full data, the model trained on REVO-LION can achieve performance comparable to simply adding all VLIT datasets up. In addition to developing an all-powerful model, REVO-LION also includes an evaluation set, which is expected to serve as a convenient evaluation benchmark for future research.

## 1 Introduction

The large-scale multimodal model GPT-4 (OpenAI, 2023) has recently exhibited strong power in generating desired answers from given images and instructions. Inspired by its remarkable success, various multimodal instruction tuning models (Chen et al., 2023; Dai et al., 2023; Li et al., 2023a; Luo et al., 2023) have been proposed towards different aspects of Vision-Language (VL) understanding, such as MiniGPT4 (Zhu et al., 2023) for detailed description and LLaVAR (Zhang et al., 2023) for text-rich image understanding. With the rapid development of Vision-Language Instruction-Tuning (VLIT), evaluating these models becomes an urgent requirement, for which several benchmarks (Liu et al., 2023c; Xu et al., 2023; Yin et al., 2023) are correspondingly proposed.

Different from existing benchmarks that concentrate on evaluating VLIT models directly, our first goal is **evaluating VLIT datasets**. The motivation comes from the insights into current VLIT models, including two similarities and one difference. *The first similarity is the model architecture*. As shown in Fig. 1, the image feature is firstly extracted by a frozen vision encoder (Fang et al., 2023). Then, a learnable projection module, which can be simply designed as the linear layer in LLaVA (Liu et al., 2023b) or the more sophisticated Q-Former in InstructBLIP (Dai et al., 2023), transfers the image feature to the language space. Finally, by feeding the transformed image feature and instruction text into the frozen Large Language Models (LLMs) (Chiang et al., 2023; Chung et al., 2022; Touvron et al., 2023), the instruction-following answer is generated. *The second similarity is the two-stage learning strategy*. During training, common large-scale image-text pairs (Ordonez et al., 2011; Schuhmann et al., 2021; Sharma et al., 2018) are leveraged for the cross-modal feature alignment in the first stage. Then, the customized high-quality instruction data is used to train the

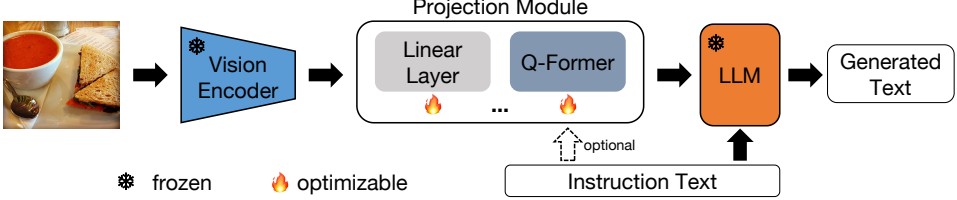

Figure 1: The common architecture in current vision-language instruction tuning methods. Extracting the visual feature by a frozen image encoder, transferring the visual feature into the language space using an optimizable projection module, generating text output via a frozen language model.

VLIT model to generate coherent and desired output in the second stage. *The difference is exact the high-quality instruction data targeting at different aspects of VL understanding*, as concluded in Table 1. To be more consistent with LLMs, the annotations in these datasets are almost generated or augmented by GPT. It follows that curating proper instruction datasets is essential in VLIT, and we thus hold that the essence of model evaluation is evaluating high-quality VLIT datasets.

For this, we pioneer a *tune-cross-evaluation* paradigm based on the common architecture in Fig. 1 to make a comprehensive analysis of VLIT datasets. Its core design is that each dataset can not only be employed for developing a model, but also can be set as a benchmark on the aspect that the dataset is constructed towards. In existing benchmarks (Liu et al., 2023c; Xu et al., 2023), the style of annotations is quite different from the style of the open-ended texts generated by LLMs, causing inappropriate assessment. Besides, human voting (Xu et al., 2023) and ChatGPT/GPT-4 (OpenAI, 2023) are proposed for performance evaluation, the former is labor-intensive and liable to cause unobjective evaluation, and the latter is inconvenient and unstable for common use because of the requirement of available API and the changeable output. Benefiting from the proposed evaluation paradigm, in which annotations consistent with LLMs are available, we define the Meta Quality (MQ) as the mean score measured by caption metrics, including BLEU, METEOR and ROUGE-L. Using MQ to measure the performance in each tune-evaluation set is more convenient and stable than GPT-involved scoring, and also more objective than human voting. Based on the proposed MQ, we devise Dataset Quality (DQ) and Sample Quality (SQ) to measure the overall capability of each dataset and sample combining all tune-evaluation sets.

Taking a step further, the other goal in this paper is **refining VLIT datasets** according to the holistic evaluation on VLIT datasets. For one thing, existing VLIT models are only equipped with one or several abilities in VL understanding, which leads to unsatisfying performance in comprehensive evaluation. For another thing, existing benchmarks build evaluation datasets by collecting datasets from different tasks (Krizhevsky et al., 2009; Lu et al., 2022) with annotations inconsistent with the open-ended generated texts (Xu et al., 2023; Yin et al., 2023), which causes inaccurate evaluation. Consequently, a dataset that integrates multiple VLIT capabilities is critical for developing an all-powerful model and building an accurate yet convenient benchmark.

To this end, we create REVO-LION by REfining VisiOn-Language InstructiOn tuNing datasets, in which samples with higher SQ from each dataset are collected. Owning fewer samples, REVO-LION is shown to be more effective than simply merging the public datasets together, which validates the effectiveness of the proposed SQ and the refinement strategy. Sufficient experiments prove that in addition to expanding the data scale, collecting high-quality data based on rational and effective evaluation is crucial for VLIT models. **We make the following main contributions:**

- We propose a *tune-cross-evaluation* paradigm on VLIT datasets. To our best knowledge, this is the first holistic analysis on VLIT datasets.
- We define the Meta Quality (MQ) as the mean score measured by BLEU, METEOR and ROUGE-L. Based on MQ, Dataset Quality (DQ) and Sample Quality (SQ) are devised to quantify the comprehensive quality of each dataset and sample in VLIT, respectively.
- We release a comprehensive dataset, namely REVO-LION, by refining public VLIT datasets. The REVO-LION includes a training and an evaluation set, the former can develop an all-powerful VLIT model, and the latter can serve as a convenient yet stable benchmark.

Table 1: The detailed comparisons of current vision-language instruction tuning datasets, including the data volume and the targets that the datasets are constructed towards.

| Datasets | Volume | Targets |
|---|---|---|
| DetGPT (Pi et al., 2023) | 5000 images and around 30000 query-answer pairs. | Reasoning-based object detection. |
| LAMM (Yin et al., 2023) | 186,098 image-language instruction-response pairs. | Daily conversation, factual knowledge reasoning, detailed description, visual task dialogue. |
| LLaVAR (Zhang et al., 2023) | 16K high-quality instruction following data. | Text-rich image understanding. |
| LLaVA (Liu et al., 2023b) | 58K in conversations, 23K in detailed description, 77K in complex reasoning. | Conversations, detailed description, complex reasoning. |
| Macaw (Lyu et al., 2023) | 69K image instances. | Human-written style text generation. |
| MiniGPT4 (Zhu et al., 2023) | Around 3500 image-text pairs. | Comprehensive image description. |
| LRV (Liu et al., 2023a) | Around 120K instances. | Robust visual instruction with mitigated hallucination issue. |

## 2 RELATED WORK

### 2.1 VISION-LANGUAGE INSTRUCTION TUNING

In the last two years, ChatGPT and InstructGPT (Ouyang et al., 2022) make great achievements in solving tasks aligned with human instructions. Inspired by this, many similar Large Language Models (LLMs) (Ding et al., 2023; Du et al., 2022; Peng et al., 2023; Zhou et al., 2023) have been proposed by fine-tuning open-source LLMs such as LLaMA (Touvron et al., 2023) and GLM (Zeng et al., 2022) using instruction data. For example, Vicuna-13B (Chiang et al., 2023) is supervised fine-tuned from LLaMA-13B (Touvron et al., 2023) using 70K shared conversations with ChatGPT from ShareGPT.com; Alpaca-7B (Taori et al., 2023) is fine-tuned from LLaMA-7B (Touvron et al., 2023) on 52K instruction-following demonstrations generated by Self-Instruct (Wang et al., 2022).

Standing on the shoulder of LLMs, many Vision-Language Instruction Tuning (VLIT) models (Li et al., 2023a; Luo et al., 2023; Su et al., 2023; Ye et al., 2023) have been proposed within a year. These models are similarly constructed by using a projection module to connect the pre-trained vision model for visual perception and the language model for text generation. The projection module is firstly trained on common image-text pairs for VL alignment, then on high quality data for instruction tuning. One of the most impactful methods is InstructBLIP (Dai et al., 2023), which is built upon the VL alignment achieved by the Q-Former in BLIP2 (Li et al., 2023b). After collecting and transforming 28 datasets from 11 tasks into instruction format, InstructBLIP (Dai et al., 2023) takes the instruction as a guidance of Q-Former to extract instruction-aware visual feature for further tuning. Similar to InstructBLIP, MiniGPT4 (Zhu et al., 2023) is firstly pre-trained on large-scale datasets (Ordonez et al., 2011; Schuhmann et al., 2021) for VL alignment, then curates around 3500 high-quality instruction data, with the assistance of ChatGPT and Vicuna (Chiang et al., 2023) targeting at comprehensive image description, for instruction tuning in the second stage. LRV (Liu et al., 2023a) constructs a dataset including both positive and negative instructions for robust tuning with mitigated hallucination issues based on MiniGPT4. Simpler than MiniGPT4, LLaVA (Liu et al., 2023b) adopts a linear layer to bridge the gap between visual and language space in the first stage using 595K image-text pairs filtered from CC3M, then, by using ChatGPT and GPT-4, 158K instruction samples including conversations, detailed description and complex reasoning have been collected in LLaVA for instruction tuning in the second stage. Similar to LLaVA (Liu et al., 2023b), DetGPT (Pi et al., 2023) collects around 30000 query-answer pairs towards reasoning-based object detection for instruction tuning in the second stage, LLaVAR (Zhang et al., 2023) enhances the text-rich image understanding ability by collecting 16K text-rich image data, Macaw (Lyu et al., 2023) builds a dataset consisting of 69K instances for human-style text generation.

To make a brief summary, existing VLIT models almost share the same model architecture and the two-stage learning strategy. The significant difference is the high-quality instruction data used in the second stage. Compared with current VLIT models targeting at certain aspects, collecting a comprehensive dataset becomes the core of developing an all-powerful VLIT model.

## 2.2 VLIT Benchmarks

With the rapid development of VLIT models, how to comprehensively and effectively evaluate these models becomes a concurrent significant problem. To this end, several benchmarks (Bitton et al., 2023; Yu et al., 2023; Zeng et al., 2023) have been proposed in the last few months. The pioneering benchmark is the LVLM-eHub (Xu et al., 2023), which evaluates VLIT models by quantifying the performance and human voting in the online arena platform. In performance quantification, they utilize 47 standard benchmarks covering 6 capabilities for evaluation, and find that image caption metrics are ineffective because the style of open-ended generated texts differs from that of annotations in the benchmarks. It is reasonable because the annotations in these benchmarks are rough, simple and outdated in the context of LLMs. Immediately after LVLM-eHub, LAMM (Yin et al., 2023) is proposed for evaluation on 9 common image tasks by collecting 11 datasets. Except for task-specific metrics, LAMM adopts GPT as a judgement for performance evaluation. However, MME (Fu et al., 2023) argues that human voting and GPT scoring bring problems of subjectivity and inaccuracy. For this, MME exams perception and cognition abilities covering 14 subtasks by manually constructing instruction-answer pairs and leads the tested models to answer "yes" or "no", which is designed for objective and accurate quantitative statistics. Nevertheless, manually collected data cannot avoid human subjectivity. Besides, such performance evaluation that heavily relies on generating "yes" or "no" is not quite reasonable, because existing VLIT models usually target at detailed tasks instead of making decisions from "yes" or "no" strictly. For fine-grained ability assessment, MMBench (Liu et al., 2023c) curates a dataset covering 20 fine-grained skills, and all instances are transformed into multi-choice problems. For robust evaluation, it employs ChatGPT for answer extraction and judgement in the proposed circular evaluation strategy, which is unable to evaluate the models directly on the generated texts, and thus causes inaccurate assessment.

In short, there are three core problems in existing benchmarks: collecting datasets with annotations consistent with open-ended generated texts for evaluation, avoiding human subjectivity in data selection and evaluation, and designing stable and convenient quantification metrics. For this, we hold that we can fully use existing VLIT datasets for both model development and evaluation, because the meticulously curated datasets are supposed to convey ideal information and ability to the models. In this way, not only the evaluation datasets with proper ground truths are collected with mitigated subjectivity, but also the caption metrics can still be employed for stable and convenient evaluation. Most importantly, based on the deep investigation in Sec. 2.1, we propose shifting the focus of model evaluation, which existing benchmarks are paying great efforts on, to dataset evaluation.

## 3 Method

### 3.1 Tune-Cross-Evaluation Paradigm

We propose the *tune-cross-evaluation* paradigm to evaluate VLIT datasets given in Table 1 with English as the primary language, and without the impact of the language differences, as shown in Fig. 2. On one side, each dataset is employed to develop a model by instruction tuning. On the other side, because these VLIT datasets are almost constructed by leveraging GPT-4 (OpenAI, 2023) or ChatGPT for text generation or augmentation, each dataset also represents a standard on the aspect that the dataset is constructed towards, by which the proper annotations consistent with open-ended generated texts are accessible. Based on the VL alignment learned in the first stage by the model with the architecture in Fig. 1, at each time, we select one from these datasets for instruction tuning, and the remaining datasets are used for inference at this time. For example, when we use DetGPT (Pi et al., 2023) for instruction tuning, the tuned model equipped with great reasoning-based object detection ability will be further tested on other datasets, and they are equipped with the abilities such as daily conversation, factual knowledge reasoning, detailed description, etc. By taking turns to cycle in this way, we finally get the comprehensive quality evaluation of each dataset and each sample. To quantify the comprehensiveness, we define the Meta Quality (MQ), Dataset Quality (DQ) and Sample Quality (SQ), and detail them in the following sections.

### 3.2 Meta Quality (MQ)

In LVLM-eHub (Xu et al., 2023), the authors hold that metrics in caption tasks are ineffective for VLIT evaluation due to the style differences between the diverse open-ended generated texts and

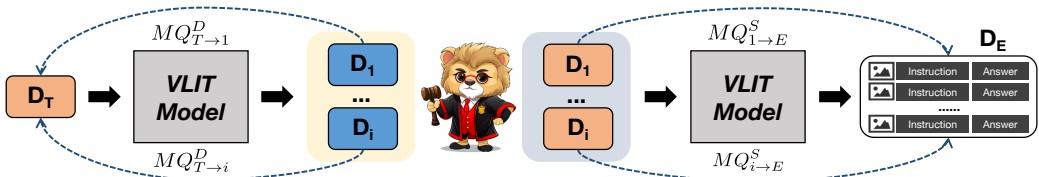

Figure 2: The overall framework of the proposed *tune-cross-evaluation* paradigm. *Left*: The diagram of Dataset Quality (DQ) evaluation. Each dataset adopted for testing measures the quality of the tuning dataset $D_T$ on the aspect that the testing datasets are constructed towards. *Right*: The diagram of Sample Quality (SQ) evaluation. Each dataset used for tuning measures how well the samples in the testing set $D_E$ match with the ability that the tuning dataset is constructed towards.

the ground-truths, which are outdated compared to LLMs. Benefiting from the proposed *tune-cross-evaluation* paradigm, when making full use of VLIT datasets as evaluation sets, the proper annotations consistent with LLMs are available. Therefore, we define the Meta Quality (MQ) as the average of scores measured by caption metrics to quantify the comprehensiveness of each dataset or sample within a single tune-evaluation experiment. Considering the time-consuming process in calculating sample-wise MQ if using SPICE, we use BLEU@1 (B@1), BLEU@2 (B@2), BLEU@3 (B@3), BLEU@4 (B@4), METEOR (M), and ROUGE-L (R) as the components for MQ definition. CIDEr is set as a hold-out metric in data refinement in Sec. 4.6. The MQ is formulated as:

$$MQ = mean(B@1 + B@2 + B@3 + B@4 + M + R). \tag{1}$$

The ablation of the combinations is studied in Sec. 4.4. It should be noted that the MQ can be commonly used to measure on a set of samples. When the number of samples is 1, it actually measures the sample-wise quality. For distinction, we denote the MQ measured on a dataset and a sample as $MQ^D$ and $MQ^S$, respectively.

### 3.3 DATASET QUALITY (DQ)

In the proposed *tune-cross-evaluation* paradigm, each time we select a dataset denoted as $D_T$ from the set of datasets $S$ for instruction tuning, the remaining datasets denoted as $D_i(i \in S, i \neq T)$ are then leveraged as evaluation ones for inference, thus measuring the quality of the tuning dataset on the aspect that the evaluation datasets are constructed towards one by one, as shown on the left side of Fig. 2. In a single tune-evaluation set, the one-side dataset quality is denoted as $MQ^D_{T \to i}$, in which the right arrow indicates the direction from the tuning dataset to the evaluation dataset. Specifically, we set the quality $MQ^D_{T \to T}$ that each tuning dataset exhibits on its aspect as 1, the maximum value of $MQ$. Therefore, when a dataset is set as the tuning one, its comprehensive quality measured by all capabilities in $S$ is formulated as the sum of one-side qualities:

$$DQ_T = MQ^D_{T \to T} + \sum_{i \in S, i \neq T} MQ^D_{T \to i} = 1 + \sum_{i \in S, i \neq T} MQ^D_{T \to i}, T \in S. \tag{2}$$

By setting each dataset as the tuning one and the remaining as evaluation ones in turn, the comprehensive DQ for all datasets can be calculated.

### 3.4 SAMPLE QUALITY (SQ)

Because the MQ can only be calculated on the inference datasets, it is hard to measure the quality of each sample in the tuning dataset when keeping the same evaluation direction in DQ, i.e., the inference datasets are regarded as standards. In contrast, when a dataset $D_E$ is set as the inference one, we hold that the model equipped with the ability of dataset $D_i(i \in S, i \neq E)$, after tuned on which, is supposed to be a standard. By this way, the $MQ^S_{i \to E}$ for each sample in $D_E$ measures how close the sample matches with the ability of the tuning dataset $D_i$, as shown on the right side of Fig. 2. To calculate the comprehensive quality that each sample exhibits on other aspects, other than DQ having the ability corresponding to itself, we define the SQ as a weighted sum:

$$SQ_E = \sum_{i \in S, i \neq E} DQ_i \cdot MQ^S_{i \to E}. \tag{3}$$

We use the $DQ_i$ as the weights for objective evaluation, the higher $DQ_i$ represents a more confident evaluation when using dataset $D_i$ to tune the model, which is regarded as a standard. By setting each dataset as the inference one and the remaining as tuning ones in turn, the comprehensive SQ for each sample exhibits on other datasets can be calculated.

## 3.5 REVO-LION

To build a comprehensive dataset integrating all capabilities of the evaluated datasets, a simple yet direct way is to merge these datasets into one without more operations. As suggested in the analysis (Zeng et al., 2023), data quality is more significant than data quantity. Therefore, we propose to REfine VisiOn-Language InstructiOn tuNing (REVO-LION) datasets according to the proposed SQ, which measures the comprehensive quality of each sample exhibits on other datasets. To preserve all capabilities, we collect samples with higher SQ from each dataset to compose REVO-LION. Formally, we denote the portion that the number of selected samples to the number of all samples in each dataset as $P$. The lower bound of SQ in dataset $D_i(i \in S)$ corresponding to the portion $P$ is $\tau_i^P$. For each sample $\boldsymbol{x}_i^k \in D_i$, if the SQ of it $SQ_i^k$ is no lower than $\tau_i^P$, the sample is collected in REVO-LION, which is formulated as:

$$S1 = \bigcup_{i \in S} \boldsymbol{x}_i^k, \quad (\boldsymbol{x}_i^k \in D_i, SQ_i^k >= \tau_i^P). \tag{4}$$

We denote this refinement strategy as S1, which is validated to be more effective than "Random Refinement" (S2) and "Gaussian Refinement" (S3) in Sec. 4.6. After creating REVO-LION from the datasets in Table 1, we split it into a training set and an evaluation set. The former can serve as a common dataset for developing an all-powerful VLIT model, and the latter can serve as a convenient benchmark covering all capabilities of these datasets and equipping with ideal annotations, based on which the caption metrics can be conveniently employed for stable evaluation.

# 4 EXPERIMENTS

## 4.1 IMPLEMENTATION DETAILS

**Data Preparation.** The evaluated VLIT datasets are clarified in Table 1. Specifically, to ensure that each dataset is independent of each other and has no overlapping samples, in DetGPT (Pi et al., 2023), we remove samples generated from MiniGPT4 (Zhu et al., 2023); in LLaVAR (Zhang et al., 2023), we remove samples generated from LLaVA (Liu et al., 2023b). Concentrating on the vision-language field, in LAMM (Yin et al., 2023) and Macaw (Lyu et al., 2023), we only use the released image-text data. In addition, the data in LLaVA (Liu et al., 2023b) is divided into three independent ones: LLaVA-Conversation (LLaVACo), LLaVA-Detailed description (LLaVADe) and LLaVA-Reasoning (LLaVARe) for their clear difference. To validate the effectiveness of the proposed data refinement strategy, we need to design an evaluation set covering all capabilities of these datasets. For this, we randomly collect $80\%$ samples from each dataset to build independent training sets, on which the *tune-cross-evaluation* paradigm and refinement are performed, and randomly collect 600 samples from the remaining $20\%$ samples to build a balanced and comprehensive evaluation set, namely "Eval600". The detailed data split process is introduced in Appendix A.1. To verify the effectiveness of the proposed evaluation paradigm and data refinement on different data partitions, we perform such data split twice and get two sets, denoted as "SPLIT1" and "SPLIT2".

**Instruction Tuning.** Due to the inferior projection ability of the linear layer adopted in LLaVA (Liu et al., 2023b) compared with the Q-Former in InstructBLIP (Dai et al., 2023) analyzed in Appendix A.3, in main experiments, we perform data evaluation and refinement using the architecture of InstructBLIP (Dai et al., 2023). The vision encoder is ViT-G/14 from EVA-CLIP (Fang et al., 2023), and the language model is Vicuna-7B (Chiang et al., 2023). More implementation details are augmented in Appendix A.2. The data evaluation and refinement experiments using the linear projection-based architecture, i.e., LLaVA (Liu et al., 2023b), are in Appendix A.3.

## 4.2 DQ EVALUATION

By setting each dataset as the tuning one $D_T$, its one-side qualities measured by other datasets $MQ_{T \to i}^D(i \neq T)$ are visualized in Fig. 3. The areas enclosed by brown and yellow lines are the

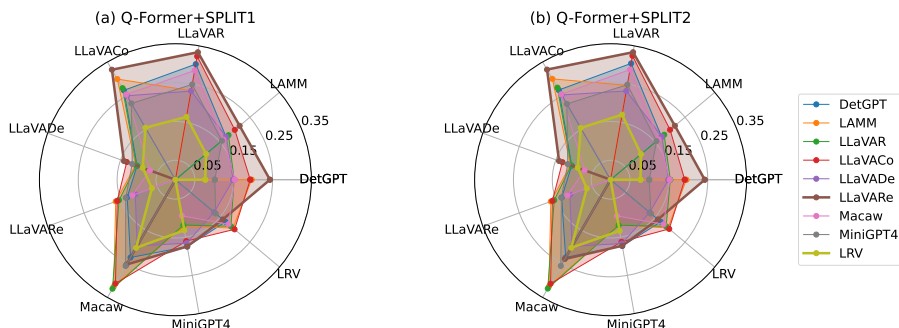

Figure 3: Visualizations of $MQ^D_{T\to i}(i \neq T)$ in dataset quality evaluation. Lines with different colors represent different datasets $D_T$ used for instruction tuning.

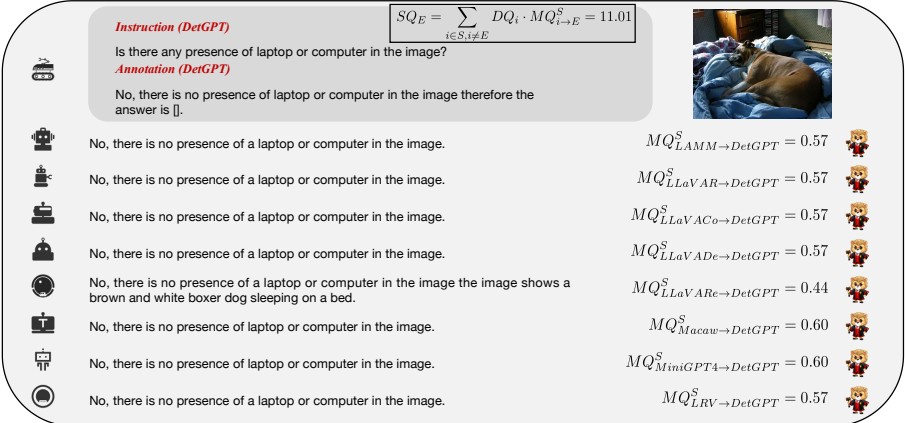

Figure 4: A sample in DetGPT with high SQ measured by other datasets.

Table 2: DQ evaluated on SPLIT1 and SPLIT2 by using the Q-Former based architecture.

| $D_T$ | DetGPT | LAMM | LLaVAR | LLaVACo | LLaVADe | LLaVARe | Macaw | MiniGPT4 | LRV |
|---|---|---|---|---|---|---|---|---|---|
| Q-Former+SPLIT1 | 2.55 | 2.63 | 2.49 | 2.68 | 2.40 | **2.85** | 2.31 | 2.38 | **1.99** |
| Q-Former+SPLIT2 | 2.56 | 2.64 | 2.50 | 2.67 | 2.41 | **2.83** | 2.32 | 2.37 | **1.99** |

largest and smallest, indicating that LLaVA-Reasoning and LRV hold the greatest and poorest comprehensive capability. It follows that LLaVA-Reasoning exhibits the highest DQ and LRV shows the lowest DQ among these datasets, as shown in Table 2 computed by Eq. 2. Besides, the results achieved on SPLIT1 and SPLIT2 demonstrate a high degree of consistency, indicating that the DQ evaluation can provide common and objective data analysis.

### 4.3 SQ Evaluation

We show a case of SQ evaluation in Fig. 4. By calculating $MQ^S_{i\to E}(i \neq E)$ between the generated answers and the annotation, the comprehensive quality of this sample can be obtained. Due to the high similarity among the generated answers, the calculated SQ of this sample is quite high. More evaluation cases of SQ are delivered in Appendix A.5.

### 4.4 Ablation Study on MQ and DQ

To validate the rationality of the definition of MQ, based on which DQ is devised, we perform ablation studies on 3 combinations of MQ, in which C1 refers Eq. 1, C2 and C3 are defined as:

$$C2 : MQ = mean(B@4 + M + R); \qquad C3 : MQ = mean(M + R). \qquad (5)$$

Table 3: Ablation study on the definition of MQ. The blue numbers after the results represent their relative rankings. The bold blue numbers indicate the inconsistent ranking relations.

| $D_T$ (Q-Former+SPLIT1) | | DetGPT | LAMM | LLaVAR | LLaVACo | LLaVADe | LLaVARe | Macaw | MiniGPT4 | LRV |
|---|---|---|---|---|---|---|---|---|---|---|
| C1 | DQ | 2.55 (4) | 2.63 (3) | 2.49 (5) | 2.68 (2) | 2.40 (6) | 2.85 (1) | 2.31 (8) | 2.38 (7) | 1.99 (9) |
| | $MQ^D_{T\to Eval600}$ | 1.37 (3) | 1.35 (4) | 1.27 (5) | 1.42 (2) | 1.16 (6) | 1.54 (1) | 1.11 (8) | 1.13 (7) | 0.78 (9) |
| C2 | DQ | 2.48 (5) | 2.61 (3) | 2.57 (4) | 2.66 (2) | 2.38 (8) | 2.73 (1) | 2.46 (6) | 2.40 (7) | 2.20 (9) |
| | $MQ^D_{T\to Eval600}$ | 0.64 (5) | 0.70 (3) | 0.69 (4) | 0.73 (1) | 0.57 (8) | 0.71 (2) | 0.63 (6) | 0.59 (7) | 0.49 (9) |
| C3 | DQ | 2.79 (6) | 2.95 (3) | 2.94 (4) | 3.00 (2) | 2.72 (8) | 3.05 (1) | 2.83 (5) | 2.72 (7) | 2.59 (9) |
| | $MQ^D_{T\to Eval600}$ | 0.50 (6) | 0.56 (3) | 0.57 (2) | 0.59 (1) | 0.47 (8) | 0.54 (4) | 0.53 (5) | 0.48 (7) | 0.44 (9) |

Table 4: $MQ^D_{T\to Eval600}$ achieved on SPLIT1 and SPLIT2 by using the Q-Former based architecture.

| $D_T$ | DetGPT | LAMM | LLaVAR | LLaVACo | LLaVADe | LLaVARe | Macaw | MiniGPT4 | LRV | Merge |
|---|---|---|---|---|---|---|---|---|---|---|
| Q-Former+SPLIT1 | 1.37 | 1.35 | 1.27 | 1.42 | 1.16 | 1.54 | 1.11 | 1.13 | 0.78 | **1.64** |
| Q-Former+SPLIT2 | 1.38 | 1.36 | 1.29 | 1.43 | 1.18 | 1.55 | 1.12 | 1.12 | 0.79 | **1.64** |

Given a dataset, its DQ quantified by a reasonable evaluation criteria should be consistent with its performance in the comprehensive evaluation. By setting "Eval600" as the comprehensive evaluation set and $MQ^D_{T\to Eval600}$ as the performance quantification, according to the three definitions, the results and relative orders of DQ and $MQ^D_{T\to Eval600}$ achieved on SPLIT1 are shown in Table 3. Compared with C3, C1 and C2 make more consistent results between DQ evaluation and $MQ^D_{T\to Eval600}$. Because a dataset owning higher DQ should exhibit better all-sided ability, and perform better in the comprehensive evaluation, C1 and C2 are more rational than C3. To preserve a more general evaluation covering as more metrics as possible, we choose C1 as the final definition of MQ, based on which DQ and SQ are devised.

## 4.5 SINGLE DATASET VS. MERGED DATASET

To build a comprehensive dataset integrating all capabilities, a simple yet direct way is to add all these single datasets together into one, denoted as "Merge". By setting "Eval600" as the evaluation dataset, the $MQ^D_{T\to Eval600}$ achieved by setting each single dataset and the merged dataset as tuning one $D_T$ is compared in Table 4. The simply merged dataset achieves the greatest result, showing adding all datasets together can contribute to an all-powerful model that exhibits the best performance on the comprehensive evaluation set covering all capabilities.

## 4.6 REVO-LION EVALUATION AND ABLATION STUDY ON REFINEMENT STRATEGY

It has been validated that combining all datasets together can develop an all-powerful model in a comprehensive evaluation compared with single datasets in Sec. 4.5. Considering that data quality is more significant than data quantity (Zeng et al., 2023), we further perform data refinement based on the above holistic evaluation. Specifically, we collect part of the samples from each dataset to build a comprehensive dataset. In addition to the refinement strategy defined in Eq. 4, denoted as S1, we design another two strategies for comparisons. The second strategy, namely S2, collects the samples from each dataset randomly with the same number as in S1. The third strategy S3 adopts the Gaussian distribution for sample selection. Specifically, for each dataset $D_i(i \in S)$, we calculate the mean value $\mu_i$ and the standard deviation $\sigma_i$ of SQ of the samples in $D_i$. The sample whose SQ exists within an interval of $\lambda$ times the standard deviation $\sigma_i$ around the mean value $\mu_i$ will be selected. S3 is formulated as:

$$S3 = \bigcup_{i\in S} \boldsymbol{x}_i^k, \quad (\boldsymbol{x}_i^k \in D_i, SQ_i^k \in [\mu_i - \lambda \cdot \sigma_i, \mu_i + \lambda \cdot \sigma_i]). \tag{6}$$

We adopt CIDEr, the hold-out metric in defining MQ, to measure the comprehensive performance of the model tuned on the refined dataset, thus making an objective evaluation for the data refinement. By setting "Eval600" as the comprehensive evaluation set, and selecting a portion $P$ of samples in each dataset, the result comparisons between S1 and S2 are given in Table 5. The case when $P = 100\%$ refers to simply adding all datasets together. For the refinement strategy S1, the re-

Table 5: Result comparisons between refinement strategies S1 and S2 with the portion $P$ ranging from 10% to 100%. "Nums" refers to the number of image-instruction-answer triplets in the refined dataset. The performance on "Eval600" is measured by CIDEr.

| Portion ($P$) | | 10% | 20% | 30% | 40% | 50% | 60% | 70% | 80% | 90% | 100% |
|---|---|---|---|---|---|---|---|---|---|---|---|
| Q-Former+SPLIT1 | Nums | 92828 | 185650 | 278473 | 371293 | 464115 | 556940 | 649760 | 742584 | 835406 | 928225 |
| | S1-CIDEr | 163.43 | 168.56 | 171.64 | 174.54 | 175.13 | 177.21 | 194.70 | 177.16 | 176.64 | 175.49 |
| | S2-CIDEr | 165.21 | 168.63 | 170.18 | 171.91 | 172.82 | 174.37 | 174.22 | 175.62 | 176.03 | 175.49 |
| Q-Former+SPLIT2 | Nums | 92807 | 185608 | 278410 | 371211 | 464012 | 556815 | 649616 | 742418 | 835219 | 928017 |
| | S1-CIDEr | 165.03 | 170.87 | 173.56 | 174.89 | 175.99 | 178.33 | 178.87 | 178.23 | 178.80 | 175.49 |
| | S2-CIDEr | 165.81 | 169.49 | 172.40 | 173.89 | 175.78 | 176.23 | 177.16 | 178.23 | 179.17 | 175.49 |

sults when $P \in [50\%, 90\%]$ are all competitive and even better than those when using the simply merged dataset. It shows that S1 successfully collects high-quality samples in the refined dataset. Specifically, the CIDEr rises with the increase of $P$ from 10% to 70%. When we select the top 50% samples with higher SQ from each dataset, we can already achieve competitive performance comparable to those using the entire data. Then, the CIDEr achieves the highest when $P = 70\%$. When $P$ increases from 70% to 100%, the CIDEr results decrease, which is caused by the involvement of samples with lower SQ. Besides, comparing S1 with S2, when keeping the number of collected samples from each dataset the same, results achieved by selecting samples with higher SQ are almost better than those achieved by random selection, which validates that S1 is more effective than S2. Moreover, for the refinement strategy S2, the CIDEr rises with the increase of $P$ from 10% to 90%. It demonstrates that with the lack of effective data evaluation and refinement strategies, a direct way for improving the performance is just expanding the scale of datasets for instruction tuning.

In addition, results from the refinement strategy S3 when setting the times $\lambda$ within [1.0, 1.5, 2.0] are given in Table 6. Comparing the results achieved by setting $\lambda = 1.0$ in S3 with those achieved by setting $P = 70\%$ in S1 in Table 5, though more samples are collected in S3, the performance achieved by S1 with fewer samples is better. The same phenomenon also happens in the comparison between setting $\lambda \in [1.5, 2.0]$ in S3 and setting $P = 90\%$ in S1 in Table 5. The comparisons prove that S1 is more effective than S3. According to the ablation studies, the effectiveness of the proposed *tune-cross-evaluation* paradigm and the refine-

Table 6: Results of data refinement using Gaussian guided strategy S3 by setting $\lambda \in [1.0, 1.5, 2.0]$. "Nums" refers to the number of image-instruction-answer triplets. The performance on "Eval600" is measured by CIDEr.

| Times ($\lambda$) | | 1.0 | 1.5 | 2.0 |
|---|---|---|---|---|
| Q-Former+SPLIT1 | Nums | 697374 | 838771 | 880426 |
| | CIDEr | 173.94 | 175.07 | 176.52 |
| Q-Former+SPLIT2 | Nums | 697346 | 838650 | 880206 |
| | CIDEr | 175.88 | 178.45 | 179.32 |

ment strategy is systematically validated. As above experiments are performed on specific data preparation for effectiveness validation, to release the REVO-LION, the evaluation and refinement are performed on the original datasets without partitions, details are given in Appendix A.4.

## 5 CONCLUSIONS AND OUTLOOK

In this paper, we are the first to perform systematic analysis of VLIT datasets and propose the *tune-cross-evaluation* paradigm. The key idea is to fully use the datasets as both tuning and evaluation sets, by which the ideal annotations consistent with open-ended generated texts are available. Then, we define the Meta Quality (MQ) as the mean score measured by caption metrics, including BLEU, METEOR and ROUGE-L. We extend MQ to Dataset Quality (DQ) and Sample Quality (SQ) for quantitative evaluation. Based on the holistic evaluation, we build a refined dataset REVO-LION by collecting samples with higher SQ from each dataset. With only half of the full data, the model trained on the refined dataset can achieve competitive performance compared to that trained on all data. In the released version, REVO-LION includes a train set, which can be commonly used for developing an all-powerful model, and an evaluation set, which can serve as a convenient yet stable benchmark. **Furthermore**, the evaluation paradigm is not limited to the datasets analyzed in this paper. The more datasets with various capabilities are involved in the evaluation, the more comprehensive analysis is achieved. As a result, the refined dataset can be used to develop a VLIT model performing well in more aspects, and also as a more comprehensive evaluation benchmark.

**Ethics Statement.** The proposed *tune-cross-evaluation* paradigm and the data refinement strategy can be generally applied on various Vision-Language Instruction Tuning (VLIT) datasets. However, a potential negative social impact is that our method is not effective in detecting non-compliant samples, such as politically sensitive and violent samples. Therefore, the refined dataset may not be able to avoid such samples, resulting in the VLIT model with the potential risk of generating non-compliant texts. Nevertheless, we build the refined dataset from the publicly released VLIT datasets without creating new samples, thus we do not introduce risks beyond these datasets.

**Reproducibility Statement.** Our work can be verified to be reproducible from the following aspects: (1) Data preparation. In Sec. 4.1, we have clarified the detailed data preparation process, including removing duplicate samples among datasets, using datasets limited to the VL fields. (2) Model selection. In Appendix A.2, we deliver the detailed setting of the models, including the architecture, the selected projection module and its pre-training data, and the hyperparameters during instruction tuning. (3) Evaluation metrics. To develop a convenient, objective and accurate benchmark, we define the Meta Quality (MQ), Dataset Quality (DQ) and Sample Quality (SQ) using caption metrics, including BLEU, METEOR and ROUGE-L. These metrics are defined by fixed rules, and the corresponding evaluation results are accurate and stable. In addition, we include the main code, the annotation file of REVO-LION-Tune and REVO-LION-Eval corresponding to Appendix A.4 in the supplementary material, which can also verify the reproducibility of our work.

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

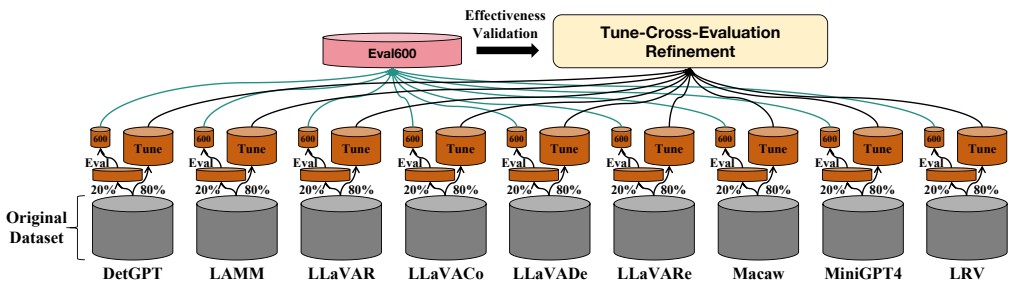

Figure 5: The diagram of the data split process. It is designed to validate the effectiveness of the proposed *tune-cross-evaluation* paradigm and the data refinement strategy in main experiments. Each original dataset is randomly divided into two parts: 80% samples are collected as a tuning set for data evaluation and refinement, and 600 samples from the remaining 20% are collected into a balanced and comprehensive evaluation set. For robust validation, we perform such partitions twice, thus creating "SPLIT1" and "SPLIT2" that are used in the main experiments.

## A  APPENDIX

### A.1  DATA SPLIT

The split for the dataset described in Sec. 4.1 (80% for training and 600 samples within the remaining 20% for testing) is used for validating the effectiveness of the proposed evaluation paradigm and the refinement strategy, as shown in Fig. 5. We choose 600 samples from each dataset for testing because the smallest dataset MiniGPT4 (Zhu et al., 2023) includes about 3,500 samples, and the 20% includes no more than 700 samples. To build a balanced and comprehensive evaluation set, we finally set the number of selected samples from each dataset for evaluation as 600.

### A.2  IMPLEMENTATION DETAILS OF INSTRUCTION TUNING

In our main experiments, we adopt the architecture of InstructBLIP (Dai et al., 2023) for data evaluation and refinement. The learnable projection module is the Q-Former in BLIP2 (Li et al., 2023b), the vision encoder is the pre-trained ViT-G/14 from EVA-CLIP (Fang et al., 2023), and the language model is Vicuna-7B (Chiang et al., 2023). Specifically, based on the selected vision encoder and language model, the Q-Former used for instruction tuning has been pre-trained on 129M images (Li et al., 2023b), including COCO (Lin et al., 2014), Visual Genome (Krishna et al., 2017), CC3M (Sharma et al., 2018), CC12M (Changpinyo et al., 2021), SBU (Ordonez et al., 2011) and LAION400M (Schuhmann et al., 2021). Based on the official code of InstructBLIP (Dai et al., 2023), the learning hyperparameters during instruction tuning are listed in Table 7. Each dataset has been adopted for tuning on 8 Nvidia A100 (80G) GPUs with the vision encoder and language model kept frozen, only parameters in the Q-Former are optimized.

In addition, we perform data evaluation and refinement in Appendix A.3 using the architecture of LLaVA (Liu et al., 2023b), which adopts the linear layer as the projection module for VL alignment. The vision encoder is the pre-trained ViT-L/14 in CLIP (Radford et al., 2021), and the language model is Vicuna-7B (Chiang et al., 2023). The linear layer used for instruction tuning has been pre-trained on 558K image-text pairs from LAION (Schuhmann et al., 2021), CC (Sharma et al., 2018) and SBU (Ordonez et al., 2011). We adopt the official code of LLaVA (Liu et al., 2023b) for instruction tuning with their default learning hyperparameters, which are given in Table 8. Each dataset has been adopted for tuning on 8 Nvidia A100 (80G) GPUs with the vision encoder and language model kept frozen, and only parameters in the linear layer are optimized.

### A.3  DATA EVALUATION AND REFINEMENT USING THE LINEAR PROJECTION MODULE

For a supplementary, we perform the data evaluation and refinement using SPLIT1 based on the architecture adopting the linear layer as the projection module for VL alignment. Specifically, we take the architecture of LLaVA (Liu et al., 2023b), and its setting has been illustrated in Appendix A.2.

Table 7: The hyperparameters for instruction tuning using the architecture of InstructBLIP (Dai et al., 2023), which adopts the Q-Former as the projection module.

| Hyperparameters | |
| --- | --- |
| Epochs | 5 |
| Warmup Epochs | 1 |
| Warmup initial learning rate | 1e-8 |
| Warmup end learning rate | 1e-5 |
| Warmup Schedule | Linear |
| Learning rate decay | Cosine |
| End (Minimum) learning rate | 0 |
| Batch size | 128 |
| Optimizer | AdamW |
| AdamW $\beta$ | (0.9, 0.999) |
| Weight decay | 0.05 |

Table 8: The hyperparameters for instruction tuning using the architecture of LLaVA (Liu et al., 2023b), which adopts the linear layer as the projection module.

| Hyperparameters | |
| --- | --- |
| Epochs | 3 |
| Learning rate | 2e-5 |
| Learning rate decay | Cosine |
| Batch size | 128 |
| Optimizer | AdamW |
| Weight decay | 0.0 |

When setting each dataset as the tuning one $D_T$, its one-side qualities $MQ_{T \to i}^{D}(i \neq T)$ measured by other datasets are given in Fig. 6 (a). It shows that each dataset exhibits extremely high similarity in the dataset-wise evaluation, leading to almost equal DQ for each dataset, compared with the results in Fig. 2. Consequently, using the linear projection-based VLIT model cannot effectively distinguish differences among datasets, resulting in invalid data evaluation. In addition, based on the evaluation, we perform the data refinement using strategies S1 and S2. The refinement results achieved by using the Q-Former and the linear layer for projection are shown in Fig. 6 (b). Obviously, when keeping both the vision encoder and the language model frozen, using the linear projection module results in a much more unsatisfying performance than using the Q-Former. Then, taking a deep comparison between the results achieved by using S1 and S2, achieved by the linear layer-based architecture, the CIDEr results vary within a small range when the portion $P$ of selected samples in each dataset changes. The highest result in S1 refinement, which is higher than using all the data, is achieved when only collecting $10\%$ of samples with higher SQ from each dataset. It shows that as a much simpler projection module, the linear layer does need as much high-quality instruction data as the Q-Former. The simplicity of the projection module limits the greatest performance that can be improved by expanding the data scale. Besides, compared with S2, the strategy S1 is almost better with different portions.

Except for the effectiveness of S1, which has been validated compared with S2, other results are inconsistent with ones using the architecture of InstructBLIP, and the linear projection module is not as good as the Q-Former. We make the deep analysis as follows. (1) From the perspective of the architecture, linear projection is quite simple in transferring the visual feature to the language space. While Q-Former adopts the pre-trained BERT (Kenton & Toutanova, 2019) as initialization, and extracts the desired visual feature using a more sophisticated cross-attention mechanism. (2) From the perspective of the pre-trained dataset, both the linear layer and the Q-Former have been pre-trained on large-scale image-text pairs for VL alignment before instruction tuning. As demonstrated in Appendix A.2, Q-Former in BLIP2 (Li et al., 2023b) has been pre-trained on 129M images (Li et al., 2023b) from COCO (Lin et al., 2014), Visual Genome (Krishna et al., 2017), CC3M (Sharma et al., 2018), CC12M (Changpinyo et al., 2021), SBU (Ordonez et al., 2011) and LAION400M (Schuhmann et al., 2021). While the linear layer in LLaVA (Liu et al., 2023b) has been pre-trained only

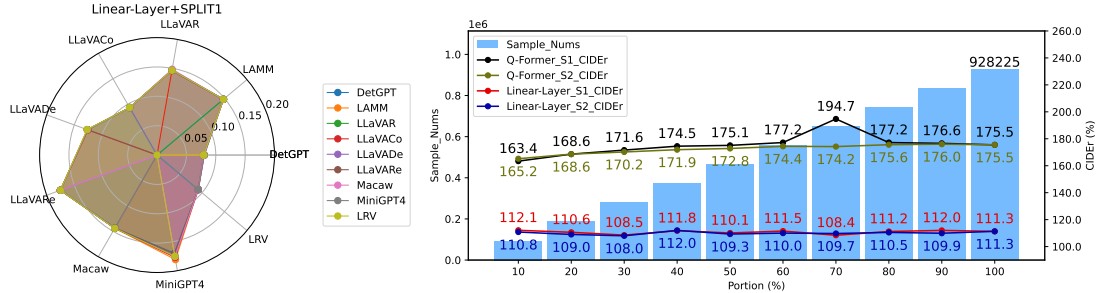

(a) Visualizations of $MQ_{T \to i}^{D}(i \neq T)$ in DQ  (b) Result comparisons between using the Q-Former and evaluation using the linear projection module. the linear layer for projection using strategies S1 and S2.

Figure 6: Results of data evaluation and refinement using the linear projection-based architecture.

Table 9: The results, measured by caption metrics, achieved by instruction tuning on REVO-LION-Tune and evaluation on REVO-LION-Eval using the Q-Former based architecture.

| Metrics | BLEU@1 | BLEU@2 | BLEU@3 | BLEU@4 | METEOR | ROUGE-L | CIDEr |
|---------|--------|--------|--------|--------|--------|---------|-------|
| Results | 0.54 | 0.43 | 0.36 | 0.31 | 0.27 | 0.53 | 2.417 |

on 558K image-text pairs from LAION (Schuhmann et al., 2021), CC (Sharma et al., 2018) and SBU (Ordonez et al., 2011). The significant difference between the amount of pre-training dataset results in a much poorer VL alignment of the linear projection than the Q-Former.

## A.4 REVO-LION RELEASE

As analyzed in Appendix. A.3, using the linear layer as the projection module for VL alignment is inferior to using the Q-Former. Therefore, we adopt the architecture of InstructBLIP, the detailed setting of which is delivered in Appendix A.2, for data evaluation and refinement. In the released dataset REVO-LION, the *tune-cross-evaluation* paradigm is directly performed on each original dataset without partition, as shown in Fig 7. According to the results in Table 5, setting the portion $P = 70\%$ can achieve the best performance. Therefore, we release the dataset with setting $P = 70\%$. After refining each dataset, we divide it into a train set for instruction tuning and an evaluation set as a convenient yet stable benchmark. To keep a balanced dataset for evaluation, we select 600 samples from each refined dataset to build the evaluation set, namely REVO-LION-Eval. The remaining samples in each refined dataset are combined into the instruction tuning dataset, namely REVO-LION-Tune. Moreover, as the annotations in REVO-LION share the same style of open-ended texts generated by LLMs, the caption metrics can be directly adopted for performance evaluation when using the REVO-LION-Eval as the benchmark.

In addition, by leveraging the architecture of InstructBLIP (Dai et al., 2023) with the settings in Appendix A.2, we perform instruction tuning using the REVO-LION-Tune, and evaluation on the REVO-LION-Eval adopting the caption metrics. The results are given in Table 9. We do not perform evaluation on REVO-LION-Eval using existing VLIT models because REVO-LION-Eval is part of the tuning datasets in developing these models. REVO-LION-Eval can be adopted as a benchmark for future research, such as architecture design.

## A.5 SQ EVALUATION CASES

Based on the proposed *tune-cross-evaluation* paradigm, for each dataset, we show several samples with high SQ and low SQ from Fig. 8 to Fig. 24.

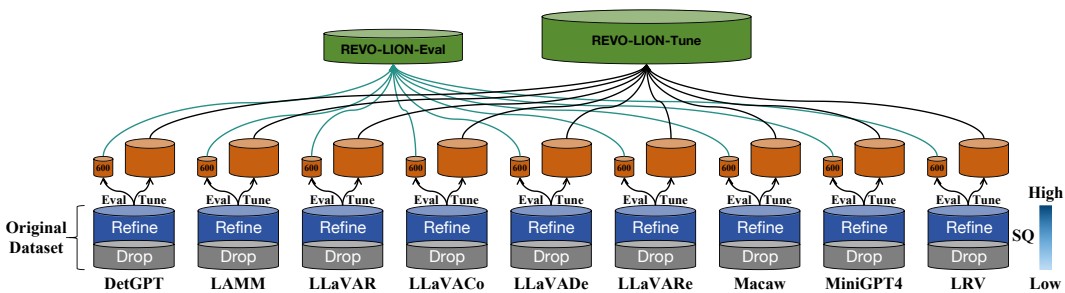

Figure 7: The refining process of buiding REVO-LION from existing VLIT datasets. The proposed *tune-cross-evaluation* paradigm is directly performed on each original dataset without partition. After the holistic evaluation, the top $70\%$ samples with higher SQ in each dataset are collected, in which 600 samples are collected into the balanced and comprehensive evaluation benchmark namely REVO-LION-Eval, and the remaining are collected into the refined tuning dataset, namely REVO-LION-Tune, for developing an all-powerful model.

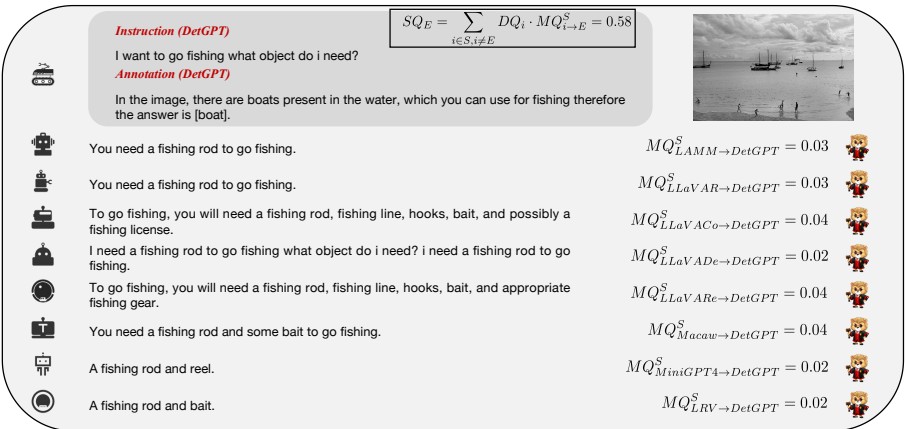

Figure 8: A sample in DetGPT with low SQ measured by other datasets.

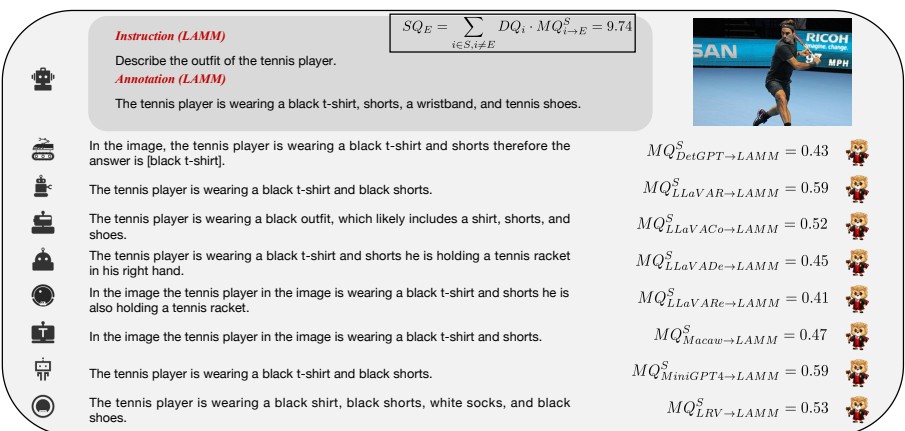

Figure 9: A sample in LAMM with high SQ measured by other datasets.

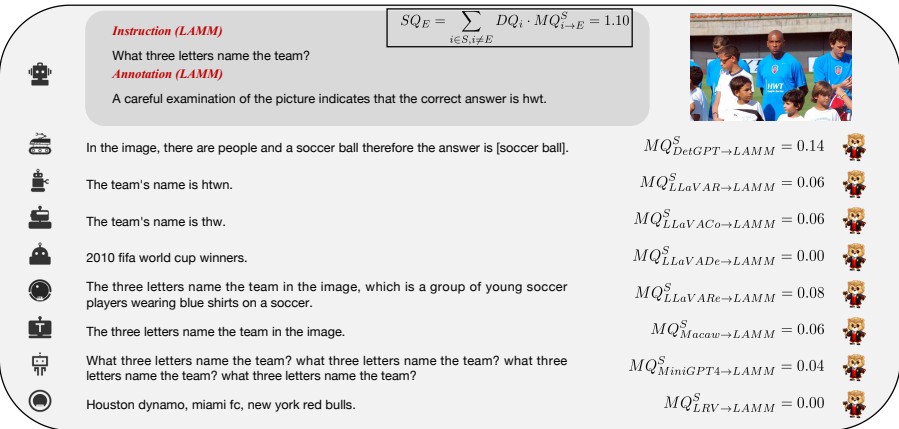

Figure 10: A sample in LAMM with low SQ measured by other datasets.

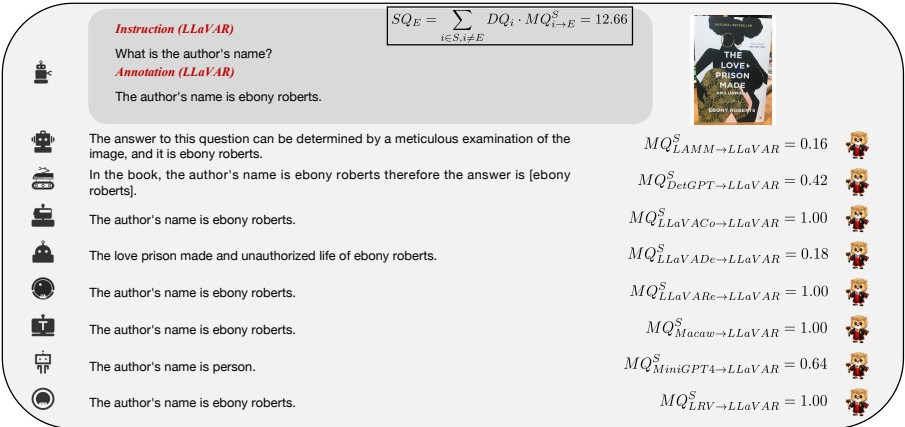

Figure 11: A sample in LLaVAR with high SQ measured by other datasets.

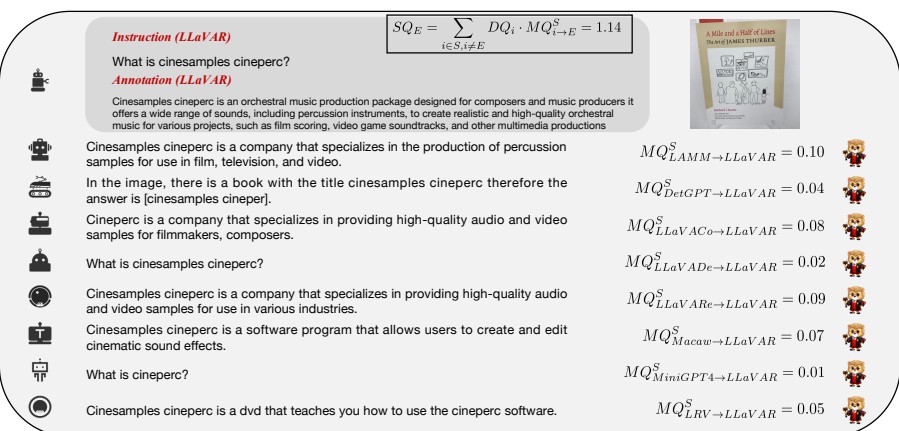

Figure 12: A sample in LLaVAR with low SQ measured by other datasets.

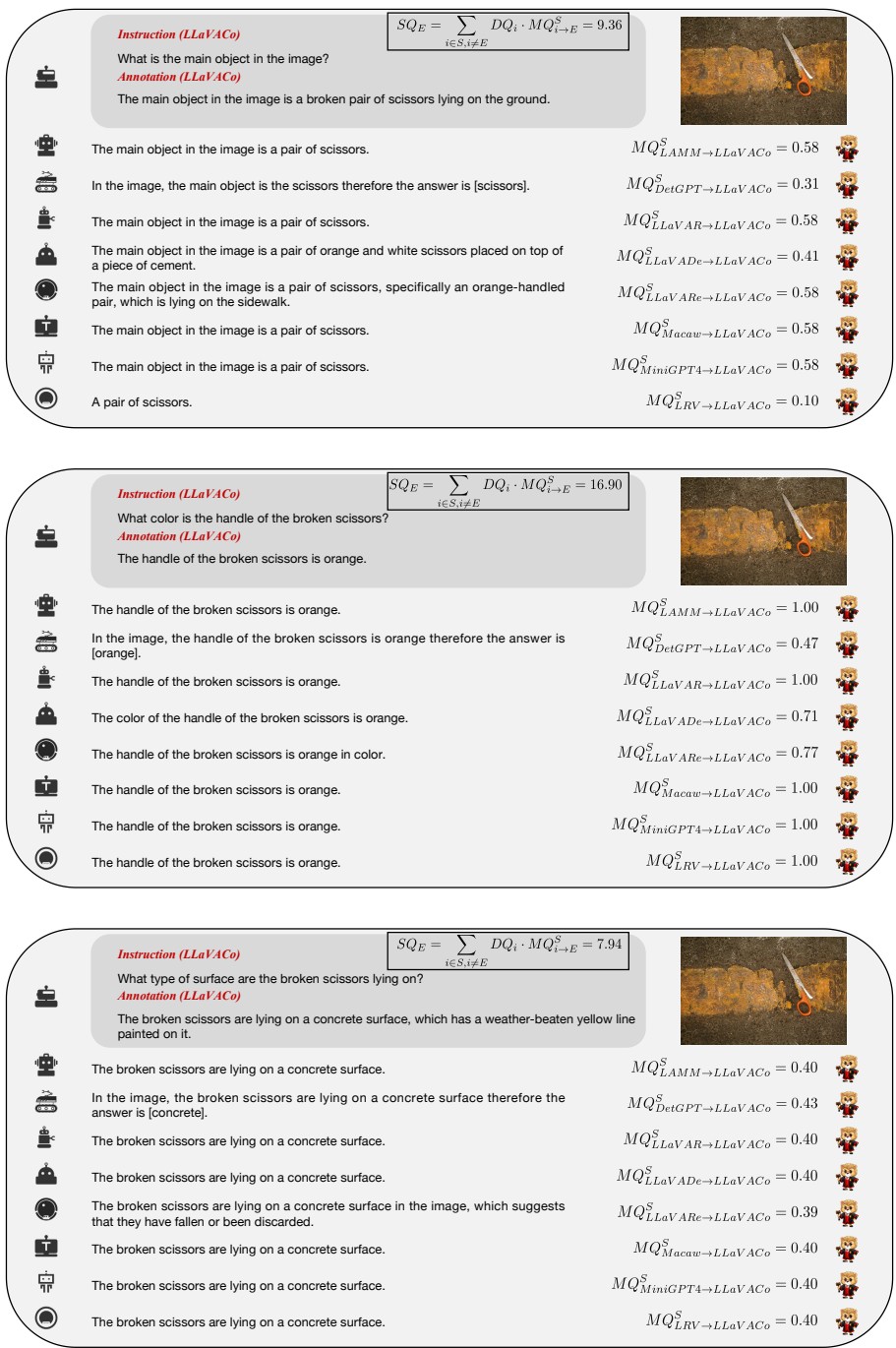

Figure 13: A conversation sample in LLaVA-Conversation with high SQ measured by other datasets.

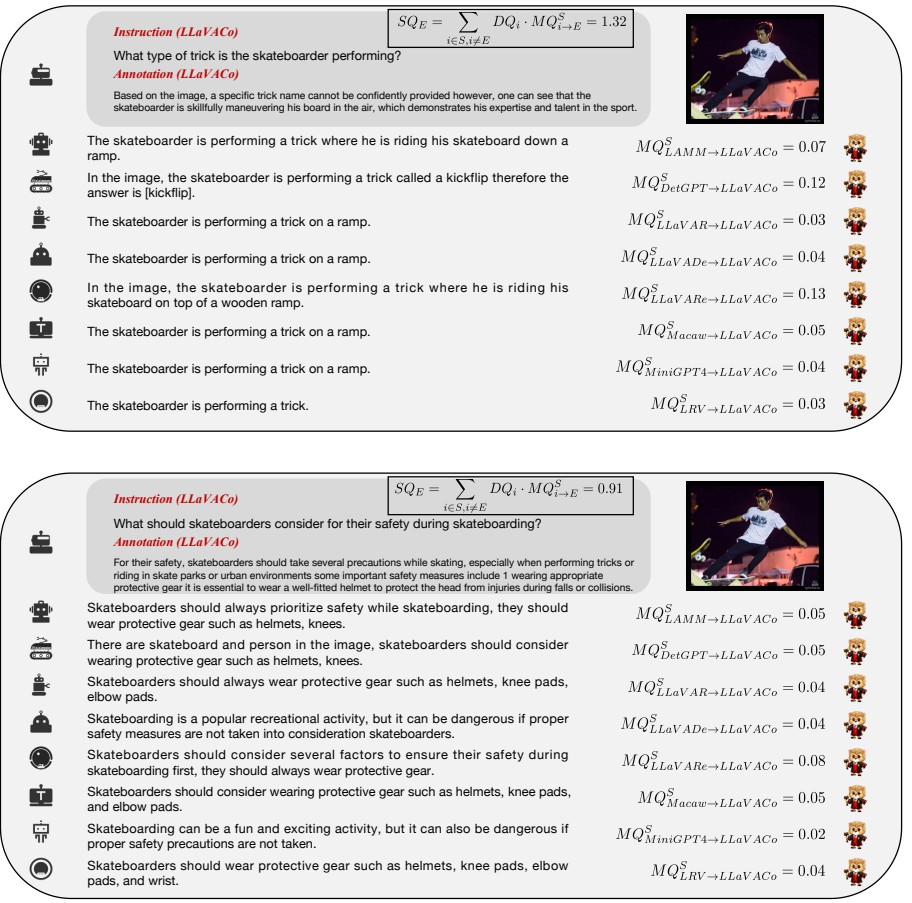

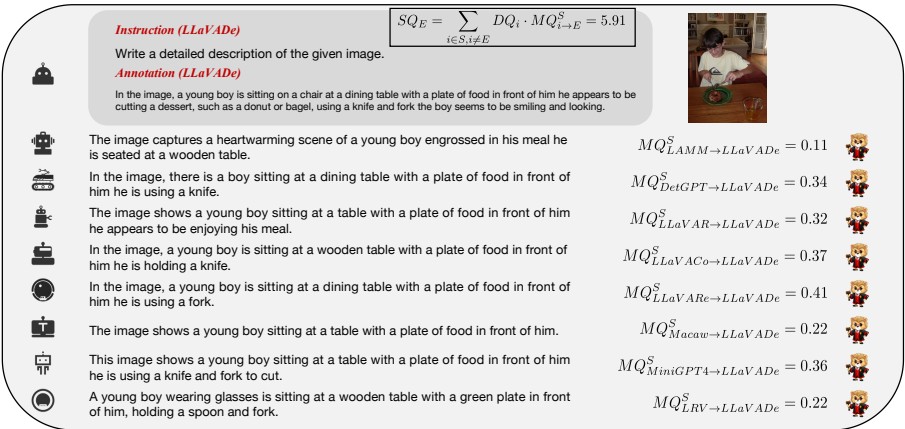

Figure 14: A conversation sample in LLaVA-Conversation with low SQ measured by other datasets.

Figure 15: A sample in LLaVA-Detailed description with high SQ measured by other datasets.

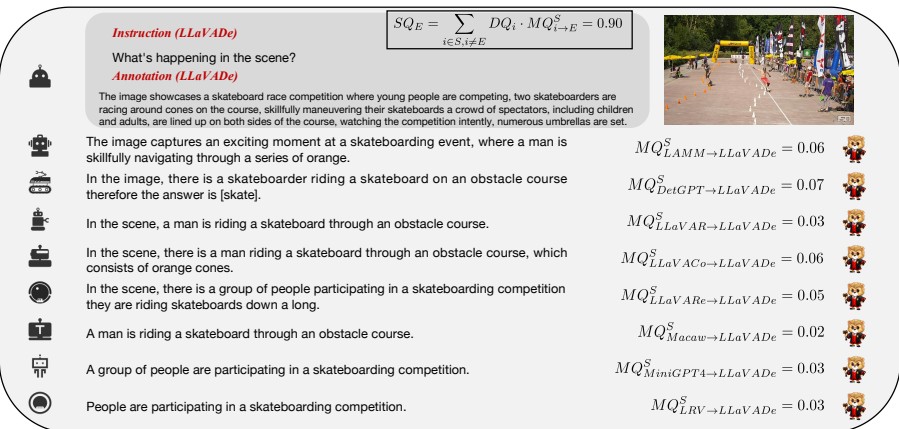

Figure 16: A sample in LLaVA-Detailed description with low SQ measured by other datasets.

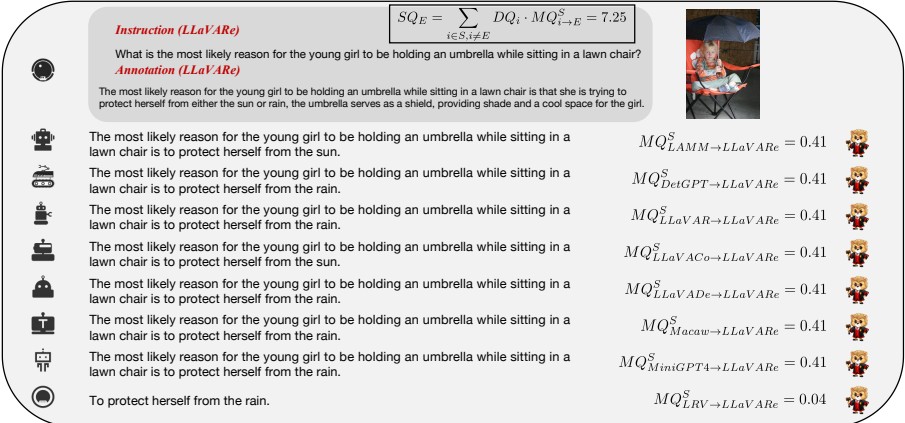

Figure 17: A sample in LLaVA-Reasoning with high SQ measured by other datasets.

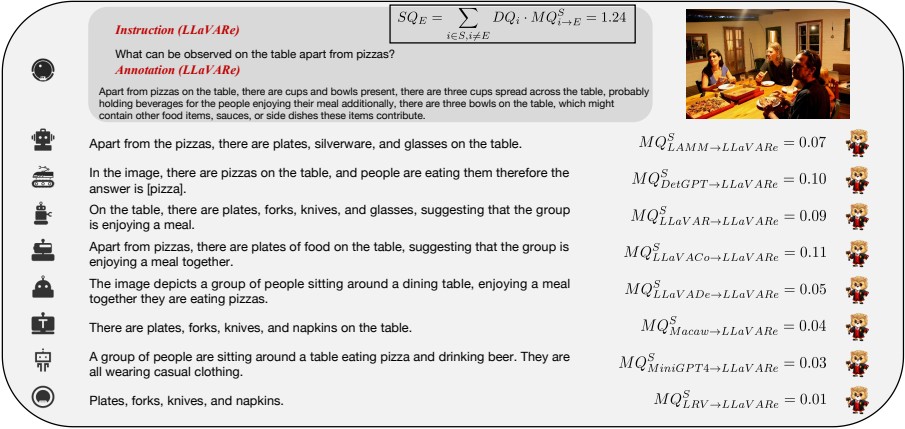

Figure 18: A sample in LLaVA-Reasoning with low SQ measured by other datasets.

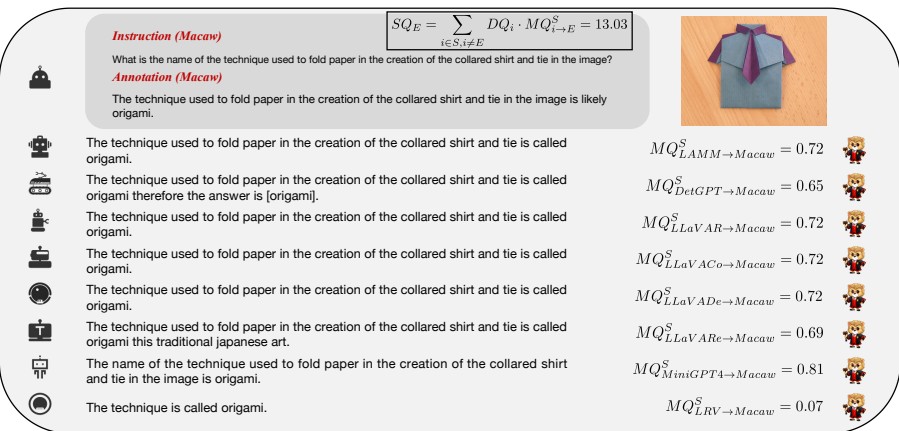

Figure 19: A sample in Macaw with high SQ measured by other datasets.

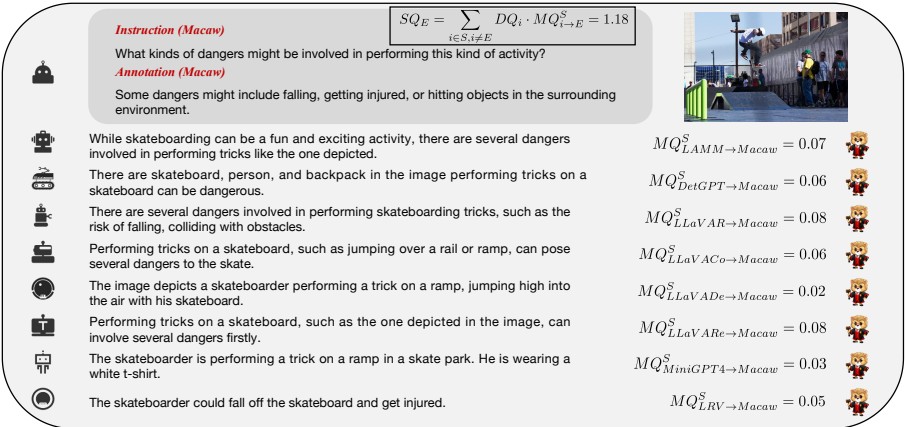

Figure 20: A sample in Macaw with low SQ measured by other datasets.

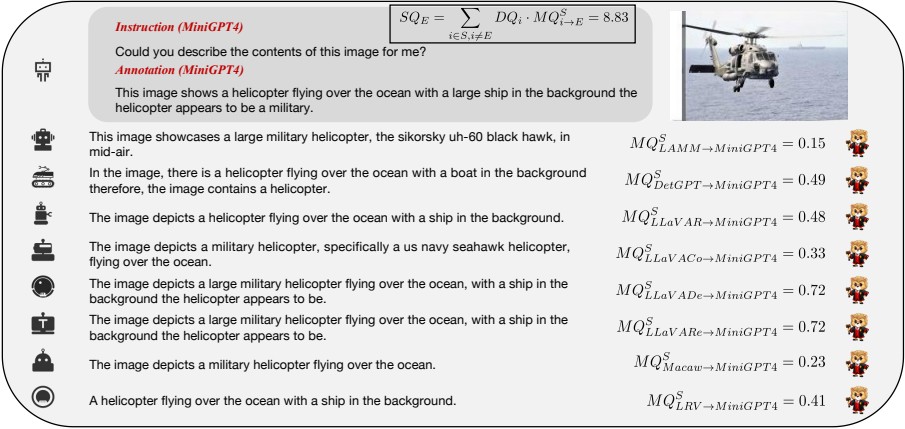

Figure 21: A sample in MiniGPT4 with high SQ measured by other datasets.

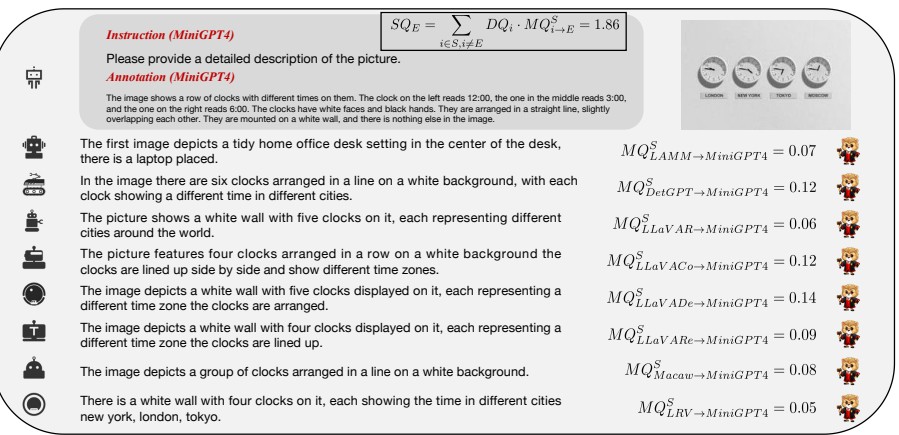

Figure 22: A sample in MiniGPT4 with low SQ measured by other datasets.

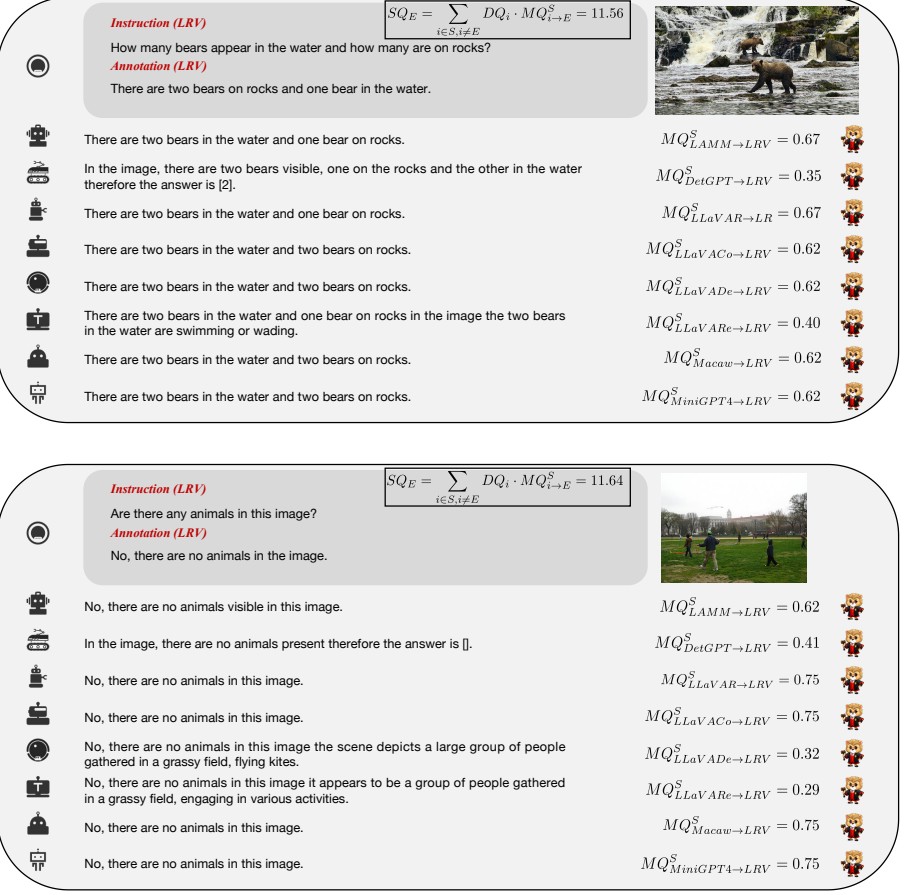

Figure 23: Two samples in LRV with high SQ measured by other datasets.

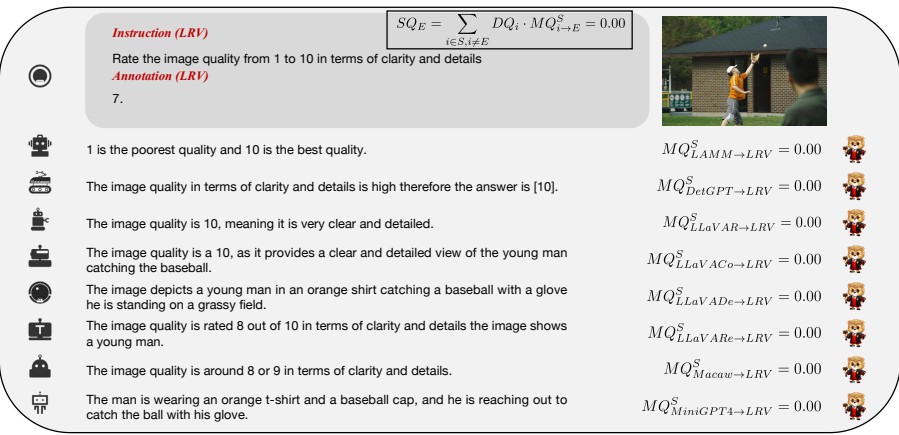

Figure 24: A sample in LRV with low SQ measured by other datasets.

