# OpenReview forum: "REVO-LION: Evaluating and Refining Vision-Language Instruction Tuning Datasets"
_ICLR.cc/2024/Conference — ICLR 2024 Conference Withdrawn Submission_

### Official Review · Reviewer_rLSJ · 2023-10-22

**Soundness:** 1 poor
**Presentation:** 2 fair
**Contribution:** 2 fair
**Rating:** 3
**Confidence:** 3

**Summary:**

This paper introduces the problem of evaluating vision-language instruction-tuning (VLIT) datasets. To this end, a **tune-cross-evaluation** paradigm is proposed: Given a collection of VLIT datasets, the quality of a particular dataset (Dataset Quality, DQ) is measured by tuning a VLIT model on this dataset and evaluating on the remaining datasets. Based on this method, this paper evaluates 9 existing VLIT datasets and reports their quality.

To improve the quality of VLIT dataset, this paper merges multiple existing VLIT datasets and select data samples based on the proposed data sample quality (SQ), which is measured by how close a sample (when used for evaluation) matches the ability of the remaining datasets for tuning. The experiments suggest that (1) the merged dataset’s quality outperforms any single dataset and (2) data samples selected by the SD metric are better than random selection.

**Strengths:**

* The idea of evaluating VLIT datasets is novel and well-motivated, which can facilitate our understanding of the quality of emerging VLIT datasets.

**Weaknesses:**

### The rationality of the proposed evaluation paradigm and metrics is questionable.
* Since most ground-truth responses of the VLIT datasets are generated by ChatGPT/GPT4 without manual verification, they are not reliable for evaluation. It is acceptable to include some noisy examples in the VLIT training set, but the quality of the evaluation dataset should be very high.
* In the **tune-cross-evaluation** setting, the tuning dataset is not included in evaluation, which may neglect a certain aspect of ability for all-powerful VLIT model. A more reasonable setting is to exclude the specific tuning datasets in evaluation but include data samples with the corresponding ability.
* According to Eq.3, SD prefers samples on which VLIT models perform well. This may exclude some noisy samples but could also exclude some challenging but meaningful samples.
* Adding the constant 1 in Eq.2 seems unnecessary.

### The experiments are insufficient to support some of the claims
* Section 2.2 argues that: (1) MME involves human subjectivity in the data collection process (2) ChatGPT-based evaluation is inaccurate, which is not supported by any evidence and there is no comparison with the proposed evaluation method in these two aspects.

**Questions:**

* How could you guarantee that existing VLIT datasets are of enough quality to serve as evaluation datasets.
* Could you provide some empirical comparison between ChatGPT-based evaluation metric and the proposed MQ that based on caption metrics?
* Could you provide some empirical evidence showing that using existing VLIT datasets alleviates human subjectivity compared with MME?

---

### Official Review · Reviewer_MuLV · 2023-10-29

**Soundness:** 3 good
**Presentation:** 3 good
**Contribution:** 3 good
**Rating:** 5
**Confidence:** 4

**Summary:**

This work tries to use systematic analysis of VLIT datasets and proposes the tune-cross-evaluation paradigm. For each tune-evaluation set, they define the Meta Quality (MQ), Dataset Quality (DQ), and Sample Quality (SQ)  to quantify the quality of a dataset or a sample, the comprehensiveness of a dataset, and the all-sided quality of each sample, respectively. Experiments show the effectiveness of the proposed method by selecting the subset of the whole dataset.

**Strengths:**

1. The motivation of the proposed method is sufficient and interesting. It could benefit current VLM research.
2. The proposed metric considers both the sample-wise and dataset-level aspects.
3. The whole method is simple with a clear presentation.

**Weaknesses:**

1. The proposed method aims to construct the evaluation metric for the vision language dataset. Therefore, an important target is to reflect the quality of each dataset and to select a high-quality subset that avoids noise and improves the performance. However, according to the experiments in Table 5, the scores of different portions are very close. And the selected split is not able to improve the whole performance. 2. Because this work claims the strategy can help performance with reduced data, it's important to compare with the original dataset that uses the same portions in Table 5. This makes the contribution more clear.
3. Whether the selected dataset using QFormer can be generalized to other frameworks, like LLaVA? This experiment should be added to show the generality of the proposed method.

**Questions:**

Please refer to the weakness section.

---

### Official Review · Reviewer_Pz2X · 2023-10-30

**Soundness:** 2 fair
**Presentation:** 2 fair
**Contribution:** 2 fair
**Rating:** 3
**Confidence:** 5

**Summary:**

The authors aim to understand the quality of the wide range of existing Vision-Language Instruction Tuning (VLIT) datasets. I believe that the problem is quite interesting and relevant to the field of generative multimodal modeling. However, I don’t think that the proposed approach is sound and well-grounded. Hence, I believe that the paper needs to revisit their dataset quality and sample quality strategy.

Comments:

- Specifically, the main strategy “tune-cross-evaluation” is based on the fact that a high-quality VLIT dataset is useful for performing well on the wide range of other VLIT datasets. This makes the comparison between the different VLIT datasets “unequal” since there is no fixed evaluation for every dataset. To support their decision, the authors mention that the existing benchmarks are ineffective in evaluating the quality of open-ended answers which I agree with. However, there are some datasets such as Visit-Bench [1] and Touchstone [2] which aim to solve this problem. If these datasets seem too recent to have been missed by the authors, I would cut some slack but would still like to see a fixed evaluation to understand what should training with the high-quality datasets be optimizing for. The authors do propose Eval600 but it is not utilized for calculating the dataset quality. It would have made more sense to me if the dataset quality was measured by fixing Eval600 as a target.

- Let us consider Llava-De (LLaVA-Detailed Description) as D_T. Intuitively, it should be a very useful dataset for VLIT training. However, it achieves a low data quality in Figure 3. This is unexpected due to the nature of the dataset quality metric (Eq. 2) which essentially penalizes the dataset for being “different” from other datasets.

- What is the intuition between Eq. 1 for Meta-quality? To the best of my understanding, Rouge-L, Meteor, and BLEU capture similar facets of lexical or span-level similarity between the two texts. Why do we need to have all of them together? In this regard, the ablation study seemed equally random and not useful. At the core, it is hard to accept a new metric without showcasing that it has high correlation with human judgments for open-ended answers. [1] shows that GPT-4 based eval has the highest agreement with humans in comparison to ROUGE, Meteor, BLEU, BertScore. For instance, the model writes a poem based on a given image and one has a ground-truth poem in the dataset. How can ROUGE quantify the usefulness of the model generated poem by just comparing it with the ground-truth one? Here, GPT-4 and Human evaluation makes more sense.


- Finally, I feel that the bi-directional nature of the quality measurements is quite confusing. Specifically, the data quality is defined based on its ability to train useful VL models (training direction), while sample quality is defined in terms of the ability of the other datasets to perform well on it (inference direction).

- I am not sure why CIDEr was considered as a heldout metric while the others were considered as in-domain metrics. I think that the authors do a poor job at providing justifications for their choices.

- I do not think that the quality of the refinement strategy is very good. Table 5 indicates the best CIDer numbers on SPLIT1 are 175.13 at 70% while 100% data gives 175.49. Is 0.36 point improvement worth going through SQ, MQ, DQ evaluations which require many extra training runs? Also I do not think the improvements are significant. On SPLIT2, we do not even observe an improvement in comparison to 100%.

References:

[1] https://arxiv.org/abs/2308.06595  \
[2] https://github.com/QwenLM/Qwen-VL/tree/master/touchstone

**Strengths:**

Mentioned in the summary

**Weaknesses:**

Mentioned in the summary

**Questions:**

Mentioned in the summary

---

### Official Review · Reviewer_u7LE · 2023-10-31

**Soundness:** 2 fair
**Presentation:** 3 good
**Contribution:** 2 fair
**Rating:** 3
**Confidence:** 3

**Summary:**

The paper proposes to evaluate the existing vision-language instruction-tuning datasets. They propose a tune-cross-evaluation paradigm which first tunes models on one dataset and then evaluate it on others in turn. Based on this paradigm, they define the Meta Quality (MQ) score which is measured by BLEU, METEOR, and ROUGE-L, based on which they propose methods to quantify the dataset or sample quality. Based on the evaluation result, they construct a dataset that consists of high-quality samples from existing datasets.

**Strengths:**

1. There are several concurrent vision-language instruction tuning datasets constructed, and it is good to compare and evaluate them in depth.
2. They demonstrate that their proposed evaluation metric can be used as a good data selection strategy.

**Weaknesses:**

1. There lacks a good meta-evaluation framework that evaluates whether their evaluation strategy is good or not. While they have demonstrated that training models with half of the data selected by their metrics can achieve comparable performance with training models with all the data, this is an indirect evaluation and it is unclear whether this means their metrics make sense, and it is hard to quantify this.
2. Their "Meta Quality" score is based on statistical evaluation metrics such as BLEU and METEOR. They have mentioned in the paper that  these metrics can be ineffective and outdated in the LLM/LVLM era, and I am not convinced that their paradigm can justify the use of such metrics for reliable evaluation.
3. As in Table 1, different datasets serve different purposes and are in different sizes, thus it is hard to use s scalar value to quantify the quality and it would be good to have a fine-grained evaluation paradigm for these datasets.

**Questions:**

How could you perform direct evaluations on your evaluation metrics?